# Design, development, and characterization of amorphous rosuvastatin calcium tablets

**Rocío González**, **Mª Ángeles Peña** *, **Norma Sofía Torres, Guillermo Torrado**

Faculty of Pharmacy, Department of Biomedical Sciences, University of Alcalá, Alcalá de Henares, Madrid, Spain

* angeles.pena@uah.es

## Abstract

This work proposes a methodology for the design, development, optimisation, and evaluation of amorphous rosuvastatin calcium tablets (BCS class II drug). The main goal was to ensure rapid disintegration and high dissolution rate of the active ingredient, thus enhancing its bioavailability. The design started from a careful selection of excipients, which due to their characteristics and proportions within the formulation allowed the use of their properties such as fluidity or granulometric distribution. The formulation was characterised using scanning electron microscopy (SEM), differential scanning calorimetry (DSC), thermogravimetry (TGA), Fourier transform infrared spectroscopy (FT-IR) and powder X-ray diffraction (PXRD) methods. The galenic SeDeM methodology was used to establish the profile of the active ingredient-excipient mixture and guarantee its suitability for producing tablets by the direct compression method. The results demonstrate that the amorphous rosuvastatin calcium tablets formulation developed made it possible to obtain cost-effective tablets by direct compression with optimal pharmacotechnical characteristics that showed a remarkable disintegration and dissolution rate. The manufactured tablets complied with the pharmacopoeia guidelines regarding content uniformity, tablet hardness, thickness, friability, in vitro disintegration time and dissolution profile.

**Data Availability Statement:** All relevant data are within the paper.

**Funding:** The author(s) received no specific funding for this work.

## 1. Introduction

Dyslipidaemias or hyperlipidaemias are alterations in the levels of lipids in the blood and are one of the main risk factors for atherosclerotic cardiovascular disease [1]. They are characterized by the presence of small dense low-density lipoprotein (LDL) particles (LDL $\geq$ 130mg/dL), hypertriglyceridaemia (TG $\geq$ 200mg/dL) and decrease in the concentration of HDL cholesterol (HDL $\geq$ 35mg/dL) [2]. The treatment includes as first option hygienic-dietary measures and glycemic control. If adequate levels are not achieved, treatment with hypolipemic drugs should be initiated, aimed at reducing LDL cholesterol, especially with statins, decreasing cardiovascular risk [3]. There are different types of statins: rosuvastatin, atorvastatin, simvastatin, pitavastatin, lovastatin, and fluvastatin. In this investigation, rosuvastatin (bis[(E)-7-[4(4-fluorophenyl)-6-isopropyl-2- [methyl(methylsulfonyl)amino] pyrimidin-5-yl] (3R,5S)-

**Competing interests:** The authors have declared that no competing interests exist.

3,5-dihydroxyhept-6-enoic acid]), a drug used to correct blood fat levels, has been selected. Rosuvastatin is a high-intensity statin and has shown pronounced results in lowering LDL [4].

Rosuvastatin is administered orally; therefore, there are multiple factors that affect the absorption of this drug, as the solubility, permeability, pH, or formulation factors. According to the Biopharmaceutical Classification System (BCS), rosuvastatin belongs to the BCS class II, having low solubility and high permeability. The dissolution and absorption rates of the drug are limited by the low solubility, this disadvantage is favored by the high permeability of the drug, which compensates to some extent for the limitations resulting from the low solubility [5]. Poor aqueous solubility is a major challenge unambiguously in the systemic delivery of orally administered BCS class II drugs. Rosuvastatin exhibits poor solubility in gastrointestinal fluids and with extensive first-pass metabolism, therefore, its oral bioavailability is limited to about 20% [6]. Different factors influence the solubility of a drug, such as shape, size and distribution of particles, surface area, porosity and surface morphology or polymorphism. Polymorphism is defined as the ability of a substance to exist in different crystalline forms in a solid state. It is a frequent phenomenon of extraordinary interest to the pharmaceutical industry [7]. Each crystalline form has a unique structure with three-dimensional order giving it its unique physical and chemical properties. In addition to the crystalline state, a drug can also exist in an amorphous state characterized by a random conformation of the molecules. The amorphous state regardless of having lower physical and chemical stability presents greater solubility than crystals, and consequently, some drugs are marketed in their amorphous form due to their greater bioavailability [8]. As a result, it is essential to understand the behaviour of the solid state and judiciously select the optimal form for galenic developments. Rosuvastatin comes in the form of a calcium salt and is presented as at least four crystalline forms (A, B, B-1, and C) and one amorphous form [9]. Form A is a pure crystalline compound; forms B and C are crystalline hydrated; and form B-1 is a dehydrated compound. When comparing the physicochemical properties of the crystalline forms, it was reported that forms B and C are much more soluble in water than form A and that this property may increase their bioavailability [10]. The amorphous form is characterized by low physical and chemical stability and high solubility [11]. As explained above despite physical and chemical instability, rosuvastatin in its amorphous state is of great interest not only because it has better bioavailability, but also because it is cheaper to manufacture compared to the crystalline form. [12].

Studies show that it is possible to increase the stability of amorphous rosuvastatin by forming a rosuvastatin free from alkali metal hygroscopic impurities, such as sodium cation [13]. In turn, the use of certain excipients influences the stability of rosuvastatin in the amorphous state [12]. The use of insoluble alkalinizing agents like calcium carbonate, calcium phosphate, calcium acetate, among others, prevents hygroscopicity and increases the intragastric pH, favoring the solubility of rosuvastatin. These alkalizing agents combined with a water-soluble diluent, such as lactose monohydrate, provide stability to the formulation. From the foregoing, the sensitivity of a drug, its stability, bioavailability, and bioequivalence in the formulation will depend directly on the polymorphic form of the drug (crystalline or amorphous) and the excipients used in the formulation [12]. Currently, more than half of the drugs are administered in the form of oral tablets [14], due to their numerous advantages over other pharmaceutical forms in term of stability, accurate dosing and ease of manufacture that ensures adherence to treatment.

Therefore, the purpose of this research is the design, development, and characterization of a galenic formulation in solid pharmaceutical dosage forms for the treatment of dyslipidaemia, choosing direct compression tablets of rosuvastatin calcium that ensure rapid disintegration and high dissolution rate of the active pharmaceutical ingredient, thus enhancing its bioavailability.

First, tablet design is started with the subsequent industrial production in mind. Therefore, the direct compression method was selected as the simplest and most cost-effective tablet manufacturing technology. At the same time, in order to achieve high dissolution rate and fast disintegration, the incorporation of superdisintegrants in the formulation is required. Therefore, it is critical a rigorous selection of excipients, considering their functionality and ratio, as well as the design of the production method that controls properties such as flow, particle size distribution or good compression.

Just before designing, developing, and optimizing a novel formulation studies were carried out on the physical and chemical compatibility of the drug and the selected excipients to meet quality standards, and thus obtain stable, safe, and effective drugs [15–17]. The physicochemical characterization of drug by Scanning Electron Microscope (SEM), Differential Scanning Calorimetry (DSC), Thermogravimetric Analysis (TGA), Fourier-transform Infrared Spectroscopy (FT-IR), and Powder X-Ray Diffraction (PXRD) were discussed. The use of spectroscopic and thermal methods in the preformulation stage is of vital importance for the detection of possible incompatibilities and physical and chemical interactions between the active ingredient and the excipients, thus selecting the most suitable excipients for the formulation design [16,18].

Finally, it was used a galenic methodology applied to studies of preformulation of tablets by direct compression, which was developed by Roig and Suñé [19–22], named the SeDeM tool, carried out with the methodology described in the Pharmacopoeia. This technique consists mainly in establishing the profile of powdered substances, drugs, excipients, or mixtures of drug plus excipients, concerning their readiness to be used in direct compression. SeDeM methodology is based on the experimental measurement of certain rheological parameters, followed by a normalisation of their values to construct the SeDeM diagram. This makes it possible to compare the results of different tests and provides the necessary information to predict whether a powder is suitable for direct compression, as well as to identify weak points that need to be corrected, allowing faster design of formulations, avoiding unnecessary studies, and reducing development time. In addition, this method facilitates robust formulation design from a scientific perspective. Finally, various features such as diameter, content uniformity, hardness, thickness, friability, disintegration time and dissolution rate were evaluated.

## 2. Materials and methods

### 2.1. Materials and reagents

Rosuvastatin calcium (Insud Pharma) (Fig 1), lactose monohydrate (Guinama), microcrystalline cellulose (Vivapur 12®, JRS Pharma GmbH& CO.KG), dibasic calcium phosphate (Emcompress®, Fagron), crospovidone (PVPP, Sigma-Aldrich) and magnesium stearate (Guinama).

### 2.2. Methodological approach

For the galenic development of the tablets, the procedure outlined in Fig 2 was carried out, starting with a search for the excipients declared by the 99 rosuvastatin formulations already on the market. Once the most frequently used excipients and their pharmacotechnical properties had been determined, the ideal excipients for cost-effective, locally manufactured formulation were defined. Therefore, a matrix was developed combining the chosen excipients, as well as the different proportions, to narrow down the number of formulations with promising technological possibilities (Stage 1, Fig 2). These formulations wee compatibility and characterization assayed in Stage 2 (Fig 2). The result of the previous stage allowed us to define the formulation that would be studied using the SeDeM methodology (Stage 3, Fig 2). Once the

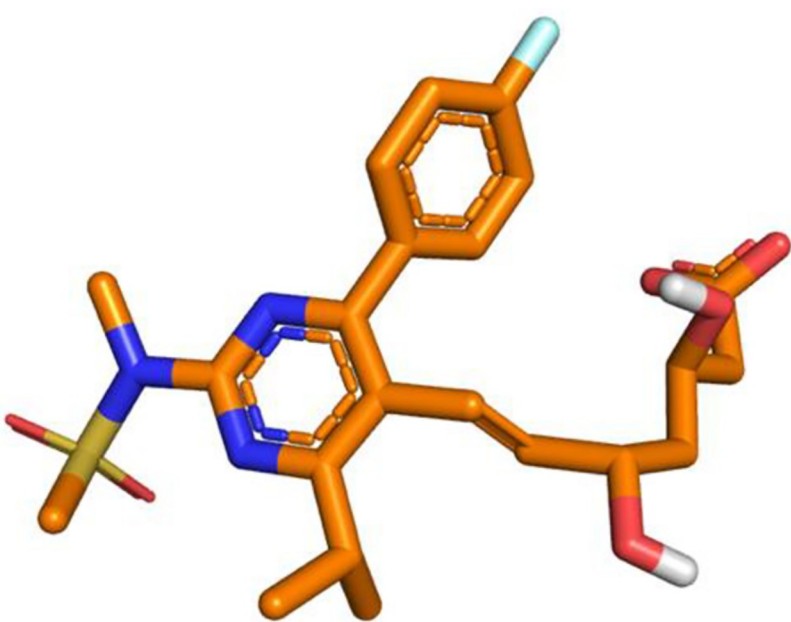

**Fig 1. Rosuvastatin 3D structure.** In orange are shown the carbon atoms, blue the nitrogen atoms, yellow the sulphur atom, in cyan the fluorine atom and the polar oxygen and hydrogen atoms in red and white, respectively.

suitability of the mixture for use in direct compression was determined, which is the most cost-effective technological method available for tablet production at the local level, the tablets were produced and characterized (Stage 4, Fig 2) to confirm the good results predicted by the methodology described.

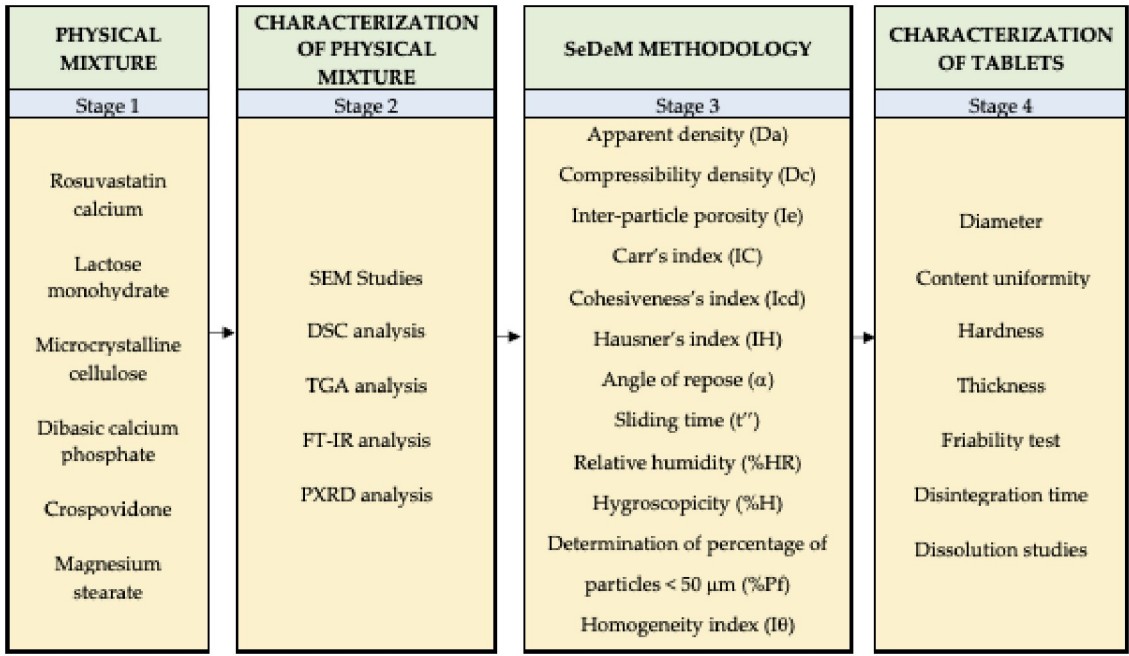

**Fig 2. Method structure of the study.**

**2.2.1. Preparation of physical mixtures.**   Physical mixtures were prepared by weighing 1:1 proportions of the active ingredient (rosuvastatin calcium) and each of the excipients (lactose monohydrate, microcrystalline cellulose, dibasic calcium phosphate, crospovidone and magnesium stearate). The components are mixed for 10 minutes in a mortar ensuring homogeneity of the mixture (Fig 2), which is stored in a desiccator protected from light and humidity until further DSC, FT-IR and PXRD analysis.

**2.2.2. Characterization of physical mixtures.**   *2.2.2.1. Scanning Electron Microscopy (SEM) Studies.* For Scanning electron microscopy (SEM) analysis, the equipment used, Zeiss DSM 959 (Germany), generates a beam of electrons (secondary electrons, SE, and backscattered electrons, BSE) with a resolution of 3 nm that impact on the sample and provide a series of signals that are registered by the different detectors of the equipment. Prior to analysis, the mixture of rosuvastatin calcium and excipients are covered with gold, making them electrically conductive.

*2.2.2.2. Differential Scanning Calorimetry (DSC) analysis.* Thermal analysis by differential scanning calorimetry (DSC) is determined with the Mettler TA 4000 DSC Star System equipment (Schwezenbach, Switzerland). The samples of approximately 3 mg weight are placed in hermetically sealed aluminium crucibles of 40–100 μL capacity. Aluminium is a totally inert material with rosuvastatin calcium and selected excipients. The samples are subjected to a constant heating rate (10 ˚C/min) under dynamic nitrogen gas purge (20 mL/min) obtaining thermograms with temperature ranges of from 30 ˚C to 400 ˚C. During the whole analysis, the melting or decomposition onset temperature is used, so that the mass does not influence the analysis. Each measurement is carried out in triplicate.

*2.2.2.3. Thermogravimetric (TGA) analysis.* The thermal stability of rosuvastatin was investigated by thermogravimetric analysis (TGA) using TGA55 (Delaware, EEUU) equipment. The experiments were carried out in aluminium crucibles (80 μL) at a heating rate of 10 ˚C per minute under nitrogen atmosphere. TGA/DTG curve was obtained in the temperature range of 25–400 ˚C on a sample of approximately 10 mg.

*2.2.2.4. Fourier transforms infrared spectroscopy (FT-IR) analysis.* The Fourier transform infrared spectroscopy (FT-IR) analysis was performed with the equipment Fourier Spectrum 2000 spectrometer Perkin Elmer® System 20000FT-IR (United States) with a resolution of 1 cm-1. For the analysis, a 1:99 dilution with KBr (sample to analyze: KBr) is homogeneously mixed in agate mortar. This mixture is placed in a hydrostatic press and by means of high pressure (5 T for 2 min), discs of about 13 mm diameter are obtained. Spectra were recorded for rosuvastatin calcium, selected excipients, and physical mixtures.

*2.2.2.5. Powder X-Ray Diffraction (PXRD) analysis.* Powder X-Ray Diffraction (PXRD) analysis was performed with the PAN-analytical multi-purpose diffractometer model X'Pert MPD. The sample is placed in a sample holder and irradiated with copper K-α radiation. The results obtained provide information about the degree of crystallinity of rosuvastatin calcium [23].

**2.2.3. SeDeM methodology.**   The SeDeM methodology was used to determine the profile of the mixture of the drug and selected excipients by evaluating a series of physical properties related to their disposition to be used in direct compression (Fig 2). The SeDeM system is based on the calculation of five Incidence Factors obtained from twelve parameters (r) of the mixture of drug and excipient [20].

*2.2.3.1. Experimental results for the SeDeM methodology*

- Apparent density (Da): Approximately 10 g of the powder mixture are weighed and poured into a test tube, noting the volume occupied in g/mL [24].

- Compressibility density (Dc): It is the volume occupied by the same amount of powder in 1, after 2500 hits on the sample in a powder density tester PT-TD200 (Germany) in g/mL [24].

- Interparticle porosity (Ie): Inter-particle porosity is calculated by means of (1), dimensionless.

$$\text{Interparticle porosity (Ie)} = \frac{(Dc - Da)}{(Dc \times Da)} \tag{1}$$

- Carr's index (IC): IC is used to calculate the compression capacity of the powder mixture in percent (2) [25].

$$\text{Carr's Index (IC)} = \frac{(Dc - Da)}{Dc} \times 100 \tag{2}$$

- Cohesiveness's index (Icd): To determine the hardness (resistance to breakage), a sample of five 120 mg tablets it is measured with a durometer Pharmatest PTB 311 (Hamburg, Germany) and the result is reported in Newtons [26].

- Hausner's index (IH): The flow and slip capacity of the powder is calculated by means of (3), which is dimensionless [25].

$$\text{Hausner's Index (IH)} = \frac{Dc}{Da} \tag{3}$$

- Angle of repose (α): A funnel 9.5 cm high, 7.2 cm in diameter of the upper mouth and 1.8 cm in diameter of the lower mouth is placed in a support at 20 cm from the surface of the test. The lower mouth of the funnel is covered, and it is filled with the powder mixture until it is flushed with the upper mouth. Then the lower mouth is uncovered to allow the powder to exit the funnel. The height of the cone (h) and the four radiuses of the base of the cone formed are measured and the average value of the radiuses (r) are calculated. The tangent of the angle of the cone formed and the value of the angle are determined by (4) [25,27].

$$\text{tg}(\alpha) = \frac{h}{r} \tag{4}$$

- Sliding time (t"): A funnel is filled with 10 g of the powder mixture and the lower mouth is covered. When the funnel is uncovered, the time it takes for all the powder to pass from the funnel to the surface is timed. If the powder does not flow, it is rated $\infty$ seconds [28,29].

- Relative humidity (%HR): The humidity is determined by applying the test of loss of mass by drying at 105.0°C ± 2.0°C during 2 h using the Rayna Liebherr Stove FKS1800 type 200041 (Germany). A sample of 4 g of powder is weighed before drying and after drying in the oven, and the different in percent is the %HR [30].

- Hygroscopicity (%H): It determines the weight increase of the sample after being kept in a humidifier at 76.0% ± 2.0% relative humidity and 22.0 °C ± 2.0°C temperature for 24 hours, the different in percent is the %H [30].

- Determination of percentage of particles < 50 μm (%Pf): 20 g of powder are weighed and then the % of powder particles passing through a 50 μm size mesh while vibrating at level three for 10 min on a vibrating shaker for cascade of CISA sieves (Barcelona, Spain) is calculated [31].

- Homogeneity index (Iθ): A sample of 50 g of powder mixture is subjected to the sieve scale with a 10-minute vibration at level three. The sieves used are 355μm, 212 μm, 100 μm,

50μμm of light placed in increasing order (5) [19,21,31,32].

$$I\theta = \frac{Fm}{100 + (dm - dm - 1) * Fm - 1 + (dm + 1 - dm) * Fm + 1 + (dm - dm - n) * Fm - n + (dm + n - dm) * Fm + n} \quad (5)$$

Fm: percentage of particles in the majority range.

Fm-1: percentage of particles in the range immediately below the majority range.

Fm+1: percentage of particles in the range immediately above the majority range.

N: order number of the fraction studied under a series, with respect to the majority fraction.

dm: mean diameter of the particles in the majority fraction.

dm-1: mean diameter of the particles in the fraction of the range immediately below the majority range.

dm+1: mean diameter of the particles in the fraction of the range immediately above the majority range.

The values obtained were the experimental results, which were used to calculate the parameters (r) using a conversion factor (Table 1). The "r parameters" are expressed into a scale from 0 to 10 and are represented graphically in the SeDeM diagram (Fig 3) [19].

Each of the 12 parameters (r) influences on some of the incidence factors that determine the suitability for use in direct compression [21,22].

*2.2.3.2. Incidence factors for the SeDeM methodology*

- Dimensional impact factor (Fdimens.): stacking capacity of the powder mixture and its effect on the tablet dimensions (6).

$$F_{dimens.} = Average(rDa; \ rDc) \quad (6)$$

- Compressibility impact factor (F$_{compressib.}$): ability of the powder mixture to be compressed and maintain its shape (7).

$$F_{compressib.} = Average(rIe; \ rIC; \ rIcd) \quad (7)$$

- Incidence factor of slippage/fluidity (F$_{flowability}$): flowability of the powder to compress (8).

$$F_{flowability} = Average(rIH; \ r\alpha; \ rt) \quad (8)$$

**Table 1. Conversion factors to obtain "r values".**

| Parameter | Conversion factor |
|---|---|
| Apparent density (**Da**) | 10 x Da |
| Compressibility density (**Dc**) | 10 x Dc |
| Interparticle porosity (**Ie**) | (10 x Ie)/1.2 |
| Carr's index (**IC**) | IC/5 |
| Cohesiveness's index (**Icd**) | Icd/20 |
| Hausner's index (**IH**) | 5 x (3 –IH) |
| Angle of repose (**α**) | 10–(α/5) |
| Slidding time (**t"**) | 10–(t"/2) |
| Relative humidity (**%HR**) | 10–%HR |
| Higroscopicity (**%H**) | 10 –(%H/2) |
| Determination of percentage of particles < 50 μm (**%Pf**) | 10 –(%Pf/5) |
| Homogeneity index (**Iθ**) | 500 x Iθ |

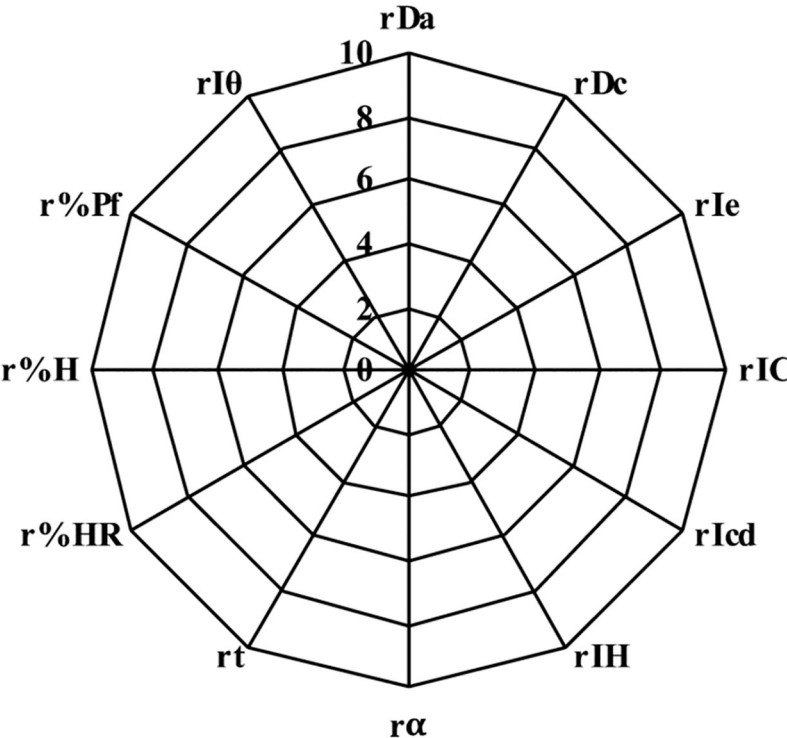

**Fig 3. SeDeM diagram for the calculated parameters (r).**

- Incidence factor of lubricity/stability ($F_{lub/stability}$): relates the residual moisture in the sample and the collection capacity to the slip and compaction capacity (9).

$$F_{lub/stability} = Average(r\%HR; \ r\%H) \tag{9}$$

- Incidence factor of lubricity/dosage ($F_{lub/dosage}$): relates the influence of the particle size distribution of the powder to the sliding capacity and the correct filling of the compression matrices (10).

$$F_{lub/dosage} = Average \ (r\%Pf; \ rI\theta) \tag{10}$$

*2.2.3.3. Incidence factors to determine the capacity to be used in direct compression for the SeDeM methodology*

From the incidence factors described above, a series of parameters are used to determine the capacity of the powder mixture to be used in direct compression [19,20]:

- Parametric index (IP) (11):

$$IP = \frac{n° \ P \geq 5}{n° \ Pt} \tag{11}$$

nº P≥5: nº of parameters (r) whose value is ≥ 5
nº Pt: nº total parameters (r) studied
The minimum expected value for a good powder mixture is IP ≥ 0.5

- Parametric profile index (IPP) (12):

$$IPP = \frac{\sum_{i=1}^{n} ri}{n} \tag{12}$$

$\sum_{i=1}^{n} ri$: sum of all parameters (r) studied
n: nº total parameters (r) studied
The minimum expected value for a powder with suitable characteristics is IPP $\geq$ 5.

- Good compression index (IGC) (13):

$$IGC = IPP \times f \tag{13}$$

IPP = parametric profile index
f = reliability factor (f = 0.952)
The minimum expected value for considering a powder suitable for direct compression is IGC $\geq$ 5.

**2.2.4. Preparation and characterization of tablets.** *2.2.4.1. Preparation of tablets formulation.* A batch of 200 tablets weighing 120 mg was manufactured. Crucial to the manufacture of the tablets are the mixing process, compression process and tablet hardness testing. For this purpose, a Powder V-blender P Prat type B nº 41412 (Barcelona, Spain) mixers (mixing process), an eccentric compression machine J. Bonals® 40B type MT (compression process) with flat face punches and a 6 mm diameter die were used. Compression was unidirectional using non-grooved punches. Pharmatest PTB 311 (Hamburg, Germany) was used to test the hardness of tablets.
*2.2.4.2. Tablets' characterization*

- Diameter and thickness
  The dimensions (diameter and thickness) of ten tablets were measured using Tablet Testing Instrument Pharmatest PTB 311 (Hamburg, Germany).

- Content uniformity
  Ten tablets from each batch were randomly weighed individually on a precision balance Mettler Toledo AG 245 (Schwerzenbach, Switzerland). The percentage of active pharmaceutical ingredient present in each dose assayed was determined and by an appropriate analytical test and in accordance with the European Pharmacopoeia the acceptance value (AV) was calculated (14) [33]. The specification establishes that the AV should be less than L1 = 15. L1 refers to the first stage of the analysis, applying the criterion of compliance with ten assayed doses.

$$AV = |M - X| + ks \tag{14}$$

X: Mean of individual contents expressed as a percentage of the label claim
M: Rerference value, in this case, 98.5.
k: Acceptability constant. In this case, 2.4.
s: Sample standard deviation.

- Hardness
  The hardness or resistance to breakage of tablets is defined as the force required to cause them to break by crushing. The equipment used is the durometer Pharmatest PTB 311 (Hamburg, Germany) which records the result in Newton. The test is carried out on 10 tablets individually.

- Friability test
  The equipment used to determine friability is the Pharmatest PTF E® (Hamburg, Germany). The analysis is performed with a sample of 20 tablets placed on a sieve to remove free dust, the sample of tablets is weighed (Pinitial) and placed inside the drum of the friabilometer and rotated 100 times at 25 rpm. Once finished, the tablets are removed, the free dust is removed, and the tablets are accurately weighed (Pfinal) [34]. From the Pinicial and Pfinal the percentage weight loss is calculated (15).

$$\% \, friability = \frac{Pinitial - Pfinal}{Pinitial} x \, 100 \qquad (15)$$

- Disintegration time
  The disintegration test is carried out on 6 tablets using a disintegration apparatus with basket assembly (Disaggregation machine Turu-Grau, Spain) with distilled water at $37.0 \pm 0.5$ ˚C. Once the apparatus is in operation, the time taken for the tablets to disintegrate completely is determined [35].

- Dissolution Studies
  The dissolution rate analysis of the tablets is performed with the Hanson Research SR8 SRII 8-Flak equipment (United States). The beaker with the 900 mL dissolution medium 0.2M phosphate buffer pH 6.8 is placed in a water bath at $37.0 \pm 0.5$˚C. The rotation speed of the paddles was $50 \pm 2$ rpm. The samples were passed through an inert filter of appropriate porosity (0.45 μm) and did not retain, to any significant extent, the dissolved active ingredient. The dissolved drugs released were analysed by UV-Vis spectrophotometry at a wavelength of 240 nm (Spectronic Helios Gamma UV-Vis Spectrophotometer, United States) [36–38]. The dissolution analysis was performed on six tablets individually.

## 3. Results and disussion

A study of drugs marketed with 20 mg of rosuvastatin prepared by direct compression considering the physicochemical and biopharmaceutical properties of the drug was performed. The selection of excipients is essential in the preparation of tablets, from their functionality to the compatibility between the drug and the excipients [17]. Due to the high number of marketed drugs, a more selective search was carried out in the first place in the CIMA database (Medicine Online Information Center of Spanish Agency of Medicines and Medical Devices) and different kinds of official books [39]. The final composition is described in Table 2.

In the formulation stage, a suitable selection of excipients and their proportions is essential, considering their functionality and technological aspects for use in direct compression [40]. Therefore, the selected excipients were diluents, binding agents, superdisintegrants and lubricants. Rosuvastatin tablets formulation was: rosuvastatin calcium (16.67%), lactose

Table 2. Composition of tablet formulation expressed in mg per tablet.

| Rosuvastatin calcium | Drug | 20.0 mg |
|---|---|---|
| Lactose monohydrate | Diluent | 68.8 mg |
| Microcrystalline cellulose (Vivapur 12®) | Diluent, binding agent, disintegrant | 12.0 mg |
| Dibasic calcium phosphate (Emcompress®) | Diluent, binding agent | 9.6 mg |
| Crospovidone (Polyvinylpolypyrrolidone, PVPP) | Disintegrant | 6.0 mg |
| Magnesium stearate | Lubricant | 3.6 mg |
| **TOTAL TABLET = 120.0 mg** | | |

monohydrate (57.33%), microcrystalline cellulose (10%), dibasic calcium phosphate (8%), crospovidone (5%) and magnesium stearate (3%). Among the selected excipients, lactose monohydrate has good flow properties but moderate compactness [40,41], so it is used in combination with excipients with good binding properties such as microcrystalline cellulose [42]. Microcrystalline cellulose (Vivapur® 12) is available in different particle sizes and moisture grades, giving it different properties and applications [43], in the present formulation, grade 12 has been chosen, as its particle size is approximately 200 μm giving it excellent flow properties. Dibasic calcium phosphate (Emcompress®) has good flow and compression properties, used as a diluent in the manufacture of these tablets, but it is necessary to incorporate a lubricating excipient such as magnesium stearate, as it is an abrasive material [43]. Magnesium stearate is incorporated because of its good non-stick properties, it prevents the formulation from sticking to the punches and the die of the compression moulding machine and it has an excellent lubricating action, which reduces the friction between the particles during the compression. The disintegration is controlled by the excipients and technological aspects of the manufacturing process [44], crospovidone as superdisintegrant was found to be ideal for rapid dispersion and for improving dissolution rate, which in turn increased the bioavailability.

The mixing process started with mixing lactose monohydrate and microcrystalline cellulose for five minutes at 30 rpm conditions. After that, dibasic calcium phosphate and rosuvastatin were added for five minutes at 30 rpm. Then, crospovidone was added for five minutes at 30 rpm and finally magnesium stearate was added for three minutes at 30 rpm. The compression process was performed with 6 mm punch at eight pressures in the bonal scale. Pharmaceutical technological characteristics were studied according to the Royal Spanish Pharmacopoeia (R. F.E.) technology test.

## 3.1. Solid-state characterization

Possible changes and interactions between rosuvastatin calcium and excipients were determined with SEM, DSC, TGA, FT-IR and PXRD, ensuring quality, security, and safety. The combination of these techniques offers interesting opportunities for the characterization of possible inconsistencies between formulation components [16].

**3.1.1. SEM studies.** Scanning Electron Microscopy (SEM) is a technique that provides topographical, structural, electrical conductivity and compositional information of the samples. Due to the high resolution that can be achieved in SEM, it is possible to know the crystalline structure, size distribution, porosity, and surface morphology of each of the excipients and drug. The SEM equipment generates an electron beam of high energy that hits the material and provides a series of signals that are registered in the different detectors of the equipment, each of which provides referenced information [45–47].

Fig 4 shows SEM studies of rosuvastatin and selected excipients in the preparation of tablets by direct compression. Rosuvastatin has irregular crystal form with flat surfaces [48]. Lactose monohydrate is a white powder with good flow properties and is widely used in combination with excipients of good binding properties, such as, microcrystalline cellulose. Vivapur 12® is presented as a white crystalline powder, composed of porous particles with a size of 180 μm, giving it excellent flow properties and good compatibility [49]. Emcompress® is composed of small-sized crystals, in direct compression it is used without grinding with physical and chemical stability, during compression the aggregates are broken, being necessary the addition of a binder that increases the particle size and therefore the specific surface without increasing the porosity, achieving good flow and compression properties [50]. Crospovidone is a white-yellow powder, in this research we have used crospovidone type A (non-micronized product) with a particle size greater than 50 μm, providing rapid disintegration and dissolution of the

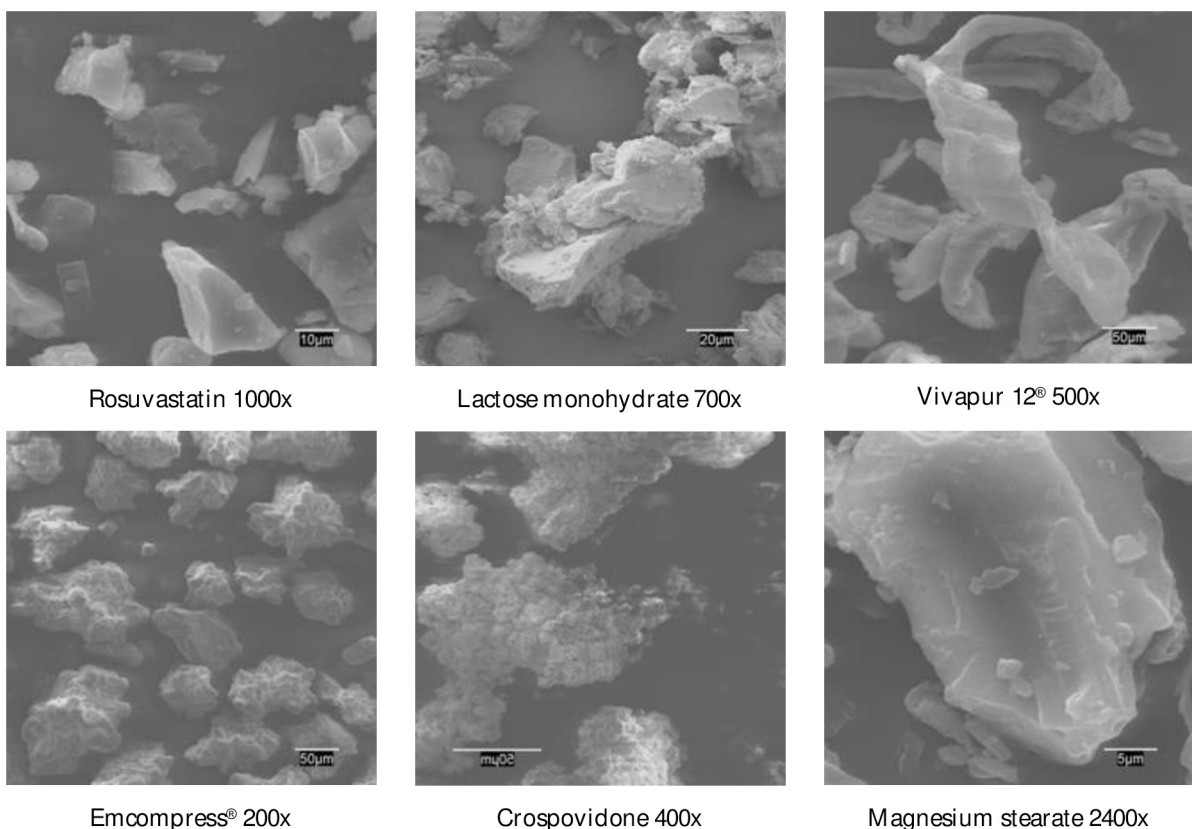

Rosuvastatin 1000x

Lactose monohydrate 700x

Vivapur 12® 500x

Emcompress® 200x

Crospovidone 400x

Magnesium stearate 2400x

**Fig 4. SEM of rosuvastatin calcium and selected excipients.**

tablets [51]. Magnesium stearate is a very fine, light white powder with very irregular edges; it is a good anti-adherent, and it has an excellent lubricating action.

**3.1.2. DSC studies.** Differential scanning calorimetry determines the amount of heat absorbed or released by a substance when it is subjected to a constant temperature and for a given time resulting in an endothermic or exothermic process. First, it was necessary to study the behavior of the API and each of the excipients individually (Fig 5). The DSC-thermogram of rosuvastatin calcium exhibits of a short endothermic peak located at Tonset = 170.15 ˚C (ΔHF = 19.80 J/g) according to the results, the amorphous form used reveals temperatures like those reported in the literature [9], preceded by an endothermic process at Tonset = 72.29 ˚C agreeing to water losses. The water content in this sample was determined by TGA analysis.

Several endothermic peaks are observed in the α-lactose monohydrate DSC. The first two at Tonset = 144.44˚C and at Tonset = 158.40˚C are a consequence of the loss of water. The third peak (Tonset = 214.45˚C) could be the anomerization of α-lactose to β-lactose due to the presence of residual water. The last peak (Tonset = 228.38˚C) is the result of fusion of crystal of lactose and subsequent decomposition. Microcrystalline cellulose (Vivapur 12®) demonstrated a first endothermic peak (Tonset = 162.86˚C) attributable to the acid hydrolysis of crystalline cellulose. However, crystallization is achieved by increasing the hydrolysis time, continuing the heating up to 301.46˚C. Calcium hydrogen phosphate (Emcompress®) reveals two endothermic peaks, the first at 110.03˚C related to the loss of water of hydration, and the second at 139.96˚C due to a crystal phase transition. As some authors affirm, dehydration occurs in two steps and depends on the variety of calcium phosphate and its particle size [52].

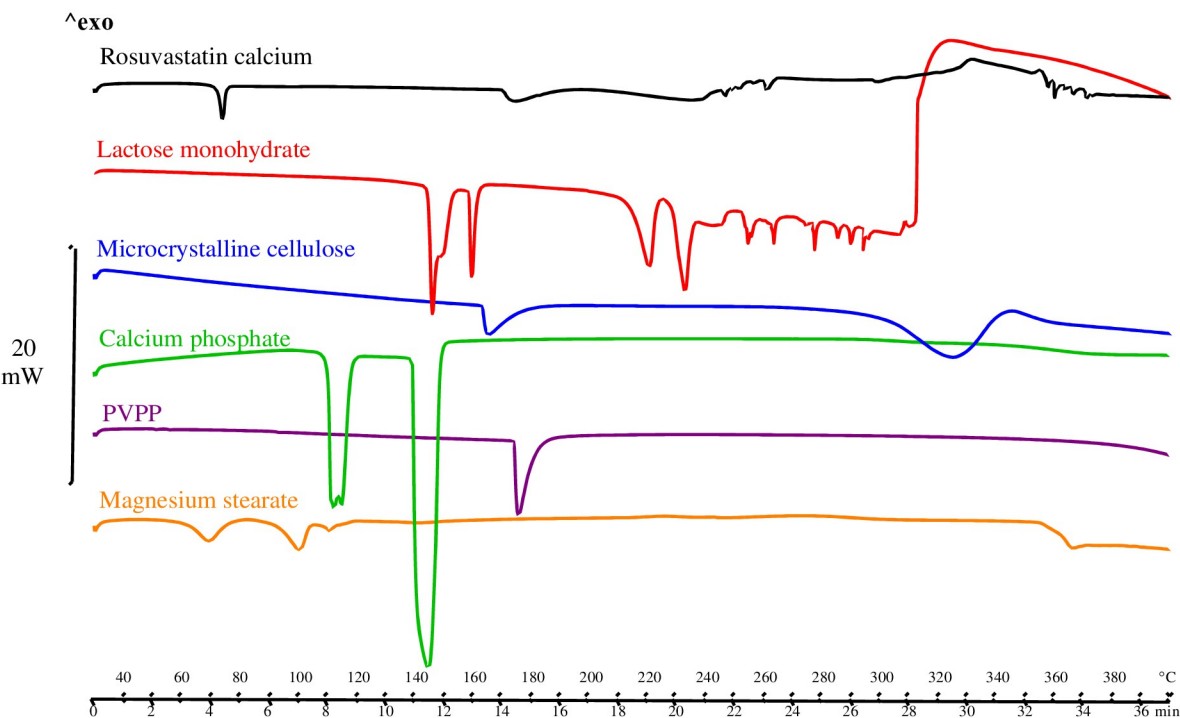

**Fig 5. DSC rosuvastatin calcium and selected excipients.**

In thermal analysis of a single endothermic peak crospovidone (Tonset = 173.91˚C) is observed consistent to its melting point, as described in the literature, PVPP has a phase glass transition (Tg) near 200.00˚C [52]. Magnesium stearate shows two first endothermic peaks corresponding to water evaporation at 61.72˚C and 93.02˚C, followed by a third peak at Ton-set = 107.97 ˚C which is due to the melting of magnesium palmitate, as stearic acid and pal-mitic acid are present in its composition (this impurity is commonly found in commercial batches).

Secondly, compatibility studies with binary mixtures of 1:1 (w/w) active substance and excipient were necessary to determine possible interactions and incompatibilities between active substance and excipient. Potential physical and chemical interactions may affect the nature, stability, and bioavailability of the drugs and, consequently, their therapeutic efficacy and safety. Shifts in the initial melting temperature, disappearance of the melting peak and/or changes in the shape of the peak may indicate physical or chemical interactions. However, it should be noted that some of these changes may be attributed to a decrease in purity or crystal-linity of the components without being an incompatibility. The results for binary (1:1) mix-tures of rosuvastatin and selected excipients are described below (Fig 6).

From what is observed in the thermogram of the 1:1 mixture of rosuvastatin and lactose monohydrate (Fig 6A), we can point out that there is a reduction of approximately 26.00˚C in the melting temperature of the drug, with similar results being observed in other drugs, such as diethylcarbamazine citrate [53]. In turn, lactose monohydrate begins dehydration at 144.44˚C, which produces a masking of the endotherm of rosuvastatin in this physical mixture. Fig 6B and 6D show the results corresponding to the physical mixtures with Vivapur 12® and PVPP, respectively. Similar results are observed, from the decrease of the temperature of melt-ing drug to overlapping endotherms in the physical mixture. In the thermogram with Vivapur 12® there is a decrease at 158.85˚C and with PVPP at 153.70˚C.

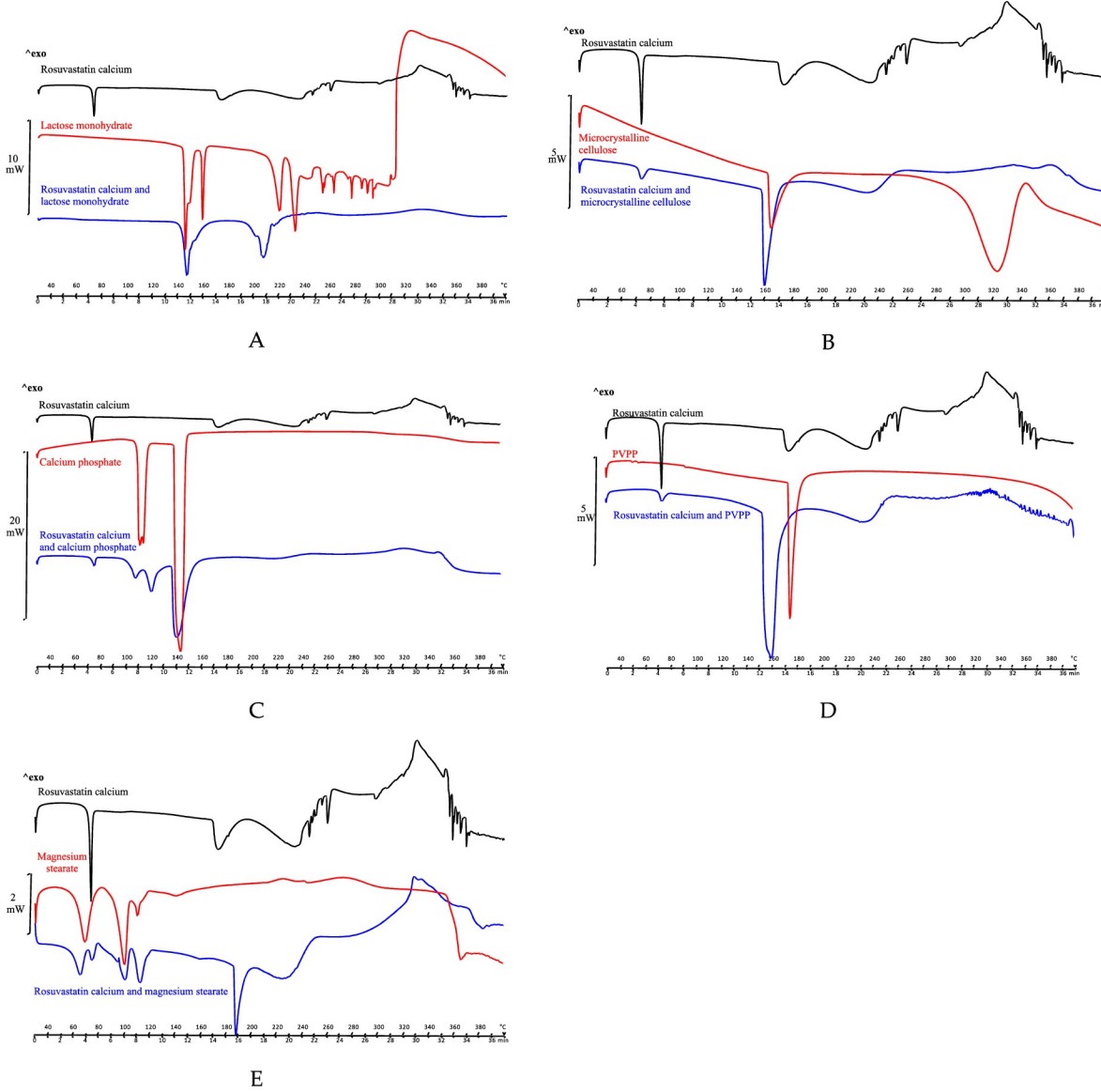

**Fig 6. DSC binary (1:1) mixtures of rosuvastatin and excipients.** (A) DSC of rosuvastatin, lactose monohydrate and physical mixture. (B) DSC of rosuvastatin, cellulose microcrystalline and physical mixture. (C) DSC of rosuvastatin, calcium phosphate and physical mixture. (D) DSC of rosuvastatin, PVPP and physical mixture. (E) DSC of rosuvastatin, magnesium stearate and physical mixture.

In the thermogram corresponding to the mixture with calcium phosphate (Fig 6C) a jump is observed at lower temperatures corresponding to the fusion of the drug, from 170.15˚C to 137.45˚C. These results are like those described in the literature [54] on the 1:1 physical mixture metronidazole and calcium phosphate, which can be attributed to a solid-solid interaction or a decrease in individual purity, but it does not necessarily indicate an incompatibility. Fig 6E, physical mixture rosuvastatin and magnesium stearate, shows a reproducibility of endothermic events with respect to each compound separately, attributing compatibility between both compounds. Small variations in relation to the melting temperature of the drug are due to the decrease in individual purity, without indicating an incompatibility.

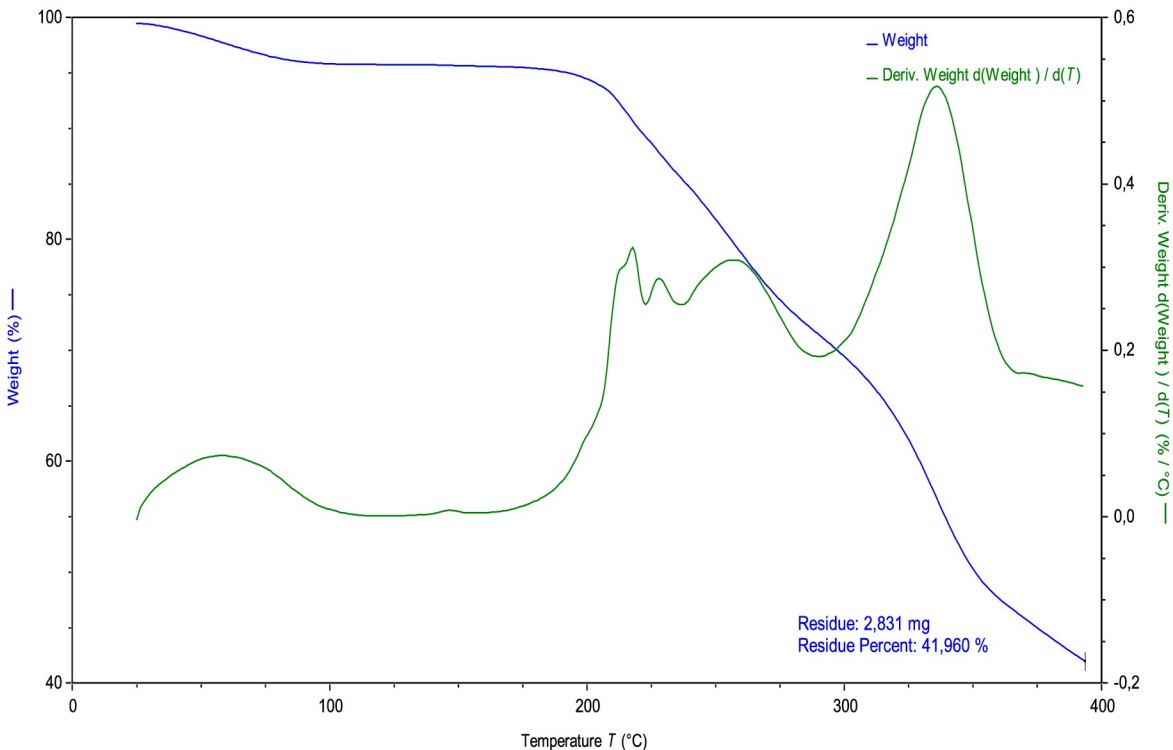

**Fig 7. TGA rosuvastatin calcium.**

**3.1.3. TGA studies.** The DSC-rosuvastatin calcium showed a hydrate, the water content (equivalent to 1 molecule of water) was determined by thermogravimetric analysis (TGA). Fig 7 shows the water loss of rosuvastatin at approximately 100.00˚C, with a weight variation of 3.7%. Following this, the drug shows stability up to 200.00˚C where the decomposition of the drug is observed with a maximum mass loss of 42.0%. These results are like those described in the literature [9].

**3.1.4. FT-IR studies.** The spectrum-rosuvastatin calcium (Fig 8) reveals certain characteristic peaks at 3376, 2932, 1437, 1336 and 1230 $cm^{-1}$ that are confirmed in the literature [55]. These peaks are representative of the presence of cyclic amines (3376 $cm^{-1}$), olefinic C-H strain of the heptane side chain (2932 $cm^{-1}$) and C = O strain of the carboxyl group (1437 $cm^{-1}$). The bands at 1336 and 1155 $cm^{-1}$ are due to symmetric and asymmetric stress vibrations due to the presence of the sulfoxide group. The peak at 1230 $cm^{-1}$ is characteristic of the C-F strain of the aromatic ring. A broad band in the 3731–3376 $cm^{-1}$ region is an indication of the presence of hydroxyl group.

An infrared comparison of the physical mixtures between API and excipients was carried out and variations were observed in the mixture with calcium phosphate (Fig 9A) and lactose monohydrate (Fig 9B), as well as variations in the thermal analysis, however, these are not changing that indicate any incompatibility between drug and excipients.

**3.1.5. PXRD studies.** PXRD studies were performed to determine the crystalline or amorphous nature of rosuvastatin calcium. The samples analyzed by powder X-ray diffraction were pure rosuvastatin calcium, mixture amorphous rosuvastatin-excipients, pulverization of tablets under study and pulverization of marketed tablets, used as a parameter of comparison and the

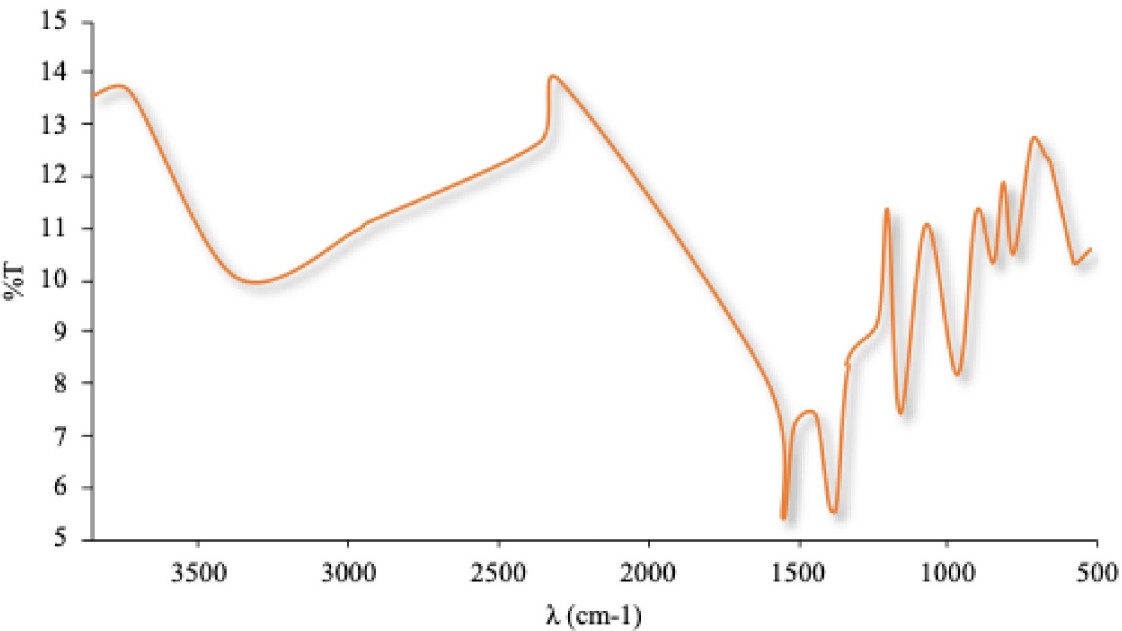

**Fig 8. FT-IR rosuvastatin calcium.**

respective PXRD patterns are given in Fig 10. The diffraction pattern of pure rosuvastatin calcium indicates amorphous nature of the drug, thus supporting the data of DSC results. In turn, we observed that the excipients used are very crystalline, both the mixture of drug and excipients and the pulverization of rosuvastatin tablets showed new peaks indicating their crystalline nature.

## 3.2. Characterization of powder blends

The suitability for direct compression tableting of rosuvastatin could be known using the SeDeM diagram method. Pharmacotechnical parameters, bulk density (rDa) and tapped density (rDc) (dimensional parameter); inter-particle porosity (rIe), Carr index (rIC) and Cohesion index (rIcd) (compressibility parameter); Hausner ratio (rIH), angle of repose (rα) and flowability (rt) (flowability parameter); loss on drying (r%HR) and hygroscopicity (r%H) (lubricity/stability parameter); and finally, particle size (r%Pf) and homogeneity index (rIθ) (lubricity/dosage parameter) (Table 3). They were experimentally determined and mathematically processed to be expressed in graphic representation as a SeDeM diagram (Fig 11). The results of this method are in line with studies of other researchers [19–22], and it is a reliable not only in preformulation studies but also in quality control tool studies and in reproducibility behaviour from batch to batch. In addition, once again it was established that the combination of some drugs compressible with suitable ingredients followed by studies SeDeM could be used as a method to identify the best excipient and calculating the maximum amount of excipients required to carry out the direct compression of a drug.

The results of this SeDeM method diagram indicate that the mixture of rosuvastatin and selected excipients (lactose monohydrate, microcrystalline cellulose, calcium phosphate, crospovidone and magnesium stearate) has indexes suitable for using in direct compression. IP above 0.5 (0.58), IPP above 5 (5.33) and good compressibility index above 5 (5.08) with bulk density (0.63 g/mL) and tapped density (0.77 g/mL) above 0.50 g/mL. At the same time, the

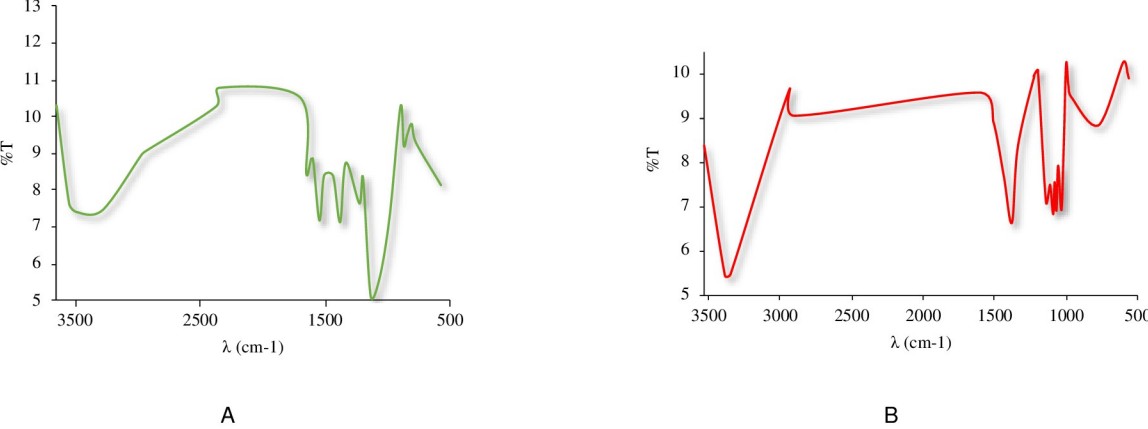

**Fig 9. FT-IR binary (1:1) mixtures of rosuvastantin and excipients.** (A) Physical mixture rosuvastatin and dibasic calcium phosphate. (B) Physical mixture rosuvastatin and lactose monohydrate.

mixture has a low percentage of particles smaller than 50 **μ**m (2.31%) and a relative humidity between 2–3% (2.61%), facilitating the compression capacity of the mixture. Despite the hygroscopicity of rosuvastatin, the mixture with the selected excipients provides a low percentage of hygroscopicity (1.36%).

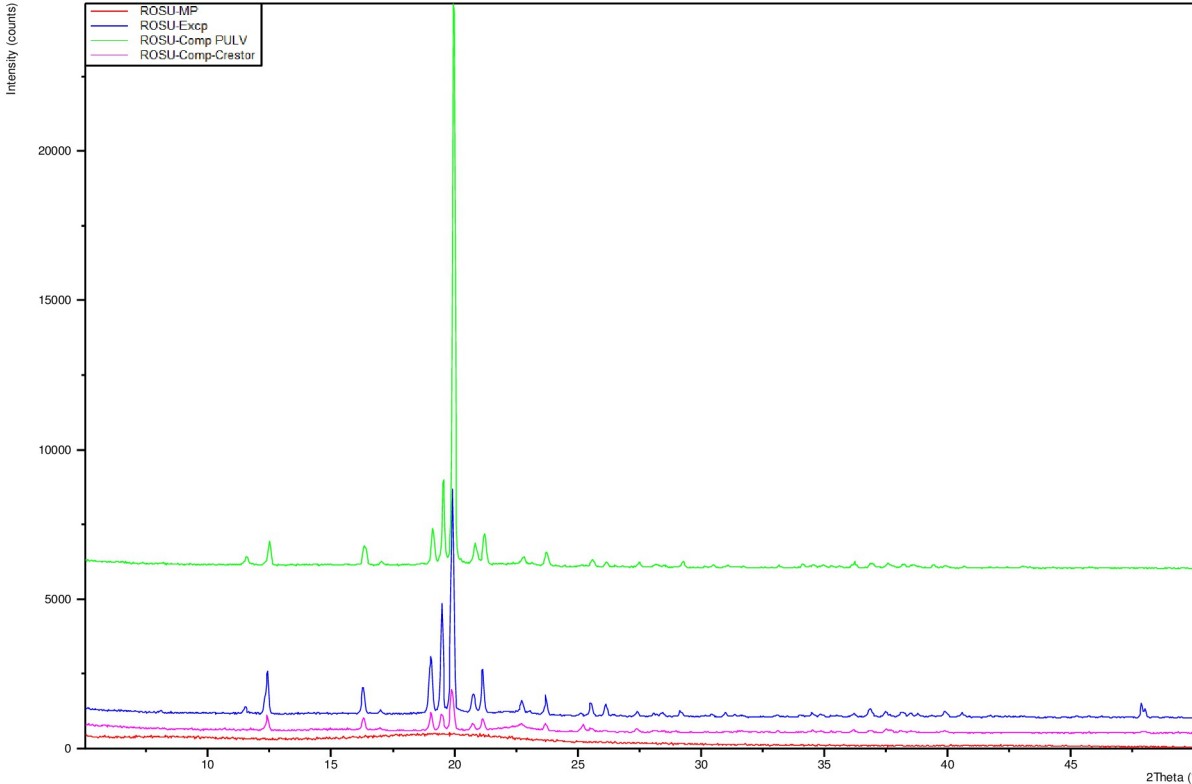

**Fig 10. PXRD rosuvastatin calcium.** In red is shown the PXRD of pure rosuvastatin calcium, pink the PXRD of marketed tablets, blue the PXRD of mixture drug-excipients, green the PXRD of tablets under study.

**Table 3. Parameters and tests used by the SeDeM method.**

| Experimental results | | | | | | | | | | | |
|---|---|---|---|---|---|---|---|---|---|---|---|
| Da (g/mL) | Dc (g/mL) | Ie | IC (%) | Icd (N) | IH | α (°) | t" | %HR | %H | %Pf | Iθ |
| 0.63 | 0.77 | 0.30 | 18.68 | 33.34 | 1.22 | 49.73˚ | ∞ | 2.61 | 1.36 | 2.31 | 0.01 |
| Parameters (r) | | | | | | | | | | | |
| rDa | rDc | rIe | rIC | rIcd | rIH | rα | rt | r%HR | r%H | r%Pf | rIθ |
| 6.27 | 7.71 | 2.48 | 3.74 | 1.67 | 8.90 | 0.05 | 0.00 | 7.39 | 9.32 | 9.54 | 6.95 |
| Impact factor | | | | | | | | | | | |
| Dimensional | | Compressibility | | Flowability | | Lubricity/ stability | | Lubricity /dosage | | | |
| 6.99 | | 2.63 | | 2.98 | | 8.36 | | 8.24 | | | |
| Index | | | | | | | | | | | |
| IP | | | IPP | | | | IGC | | | | |
| 0.58 | | | 5.33 | | | | 5.08 | | | | |

## 3.3. Pharmaceutical technological characteristics

In Table 4 are described the pharmaceutical technological characteristics of the tablets of rosuvastatin calcium obtained by direct compression, physical characteristics, dimensions, content uniformity, resistance to breaking, friability, disintegration time and dissolution rate, which must meet the requirements established according to pharmacopoeia to ensure that the quality of the tablets is as expected.

Rosuvastatin tablets had a bright white visual appearance, presented a thin thickness of 1.00 mm due to the punch used, 6 mm, and a diameter of 3.50 mm, with a breaking strength of

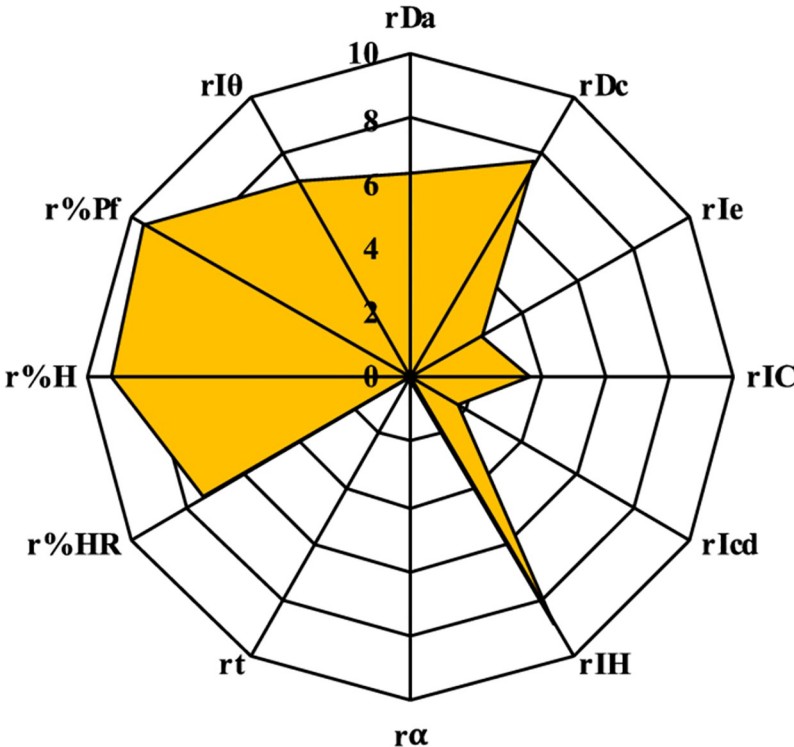

**Fig 11. SeDeM rosuvastatin for the calculated parameters (r).**

Table 4. Pharmaceutical technological characteristics of rosuvastatin tablets.

| Pharmaceutical technological characteristics | Rosuvastatin calcium tablets 20 mg |
|---|---|
| Physicals characteristics, dimension, thickness | Bright White<br>$\emptyset$ = 3.50 mm<br>T = 1.00 mm<br>(1) |
| Content uniformity | AV = 4.92<br>(2) |
| Hardness | 33.34 N |
| Friability | $W_0$ = 2.47 g<br>$W_f$ = 2.46 g<br>D = 0.40%<br>(3) |
| Disintegration Time | Between 9.00–9.48 s |
| Dissolution | Nearly 100% |

(1) Diameter ($\emptyset$) medium of 10 units. Thickness (T) medium of 10 units. (2) Acceptance value (AV). (3) $W_0$ = initial weight; $W_f$ = final weight; D = deviation.

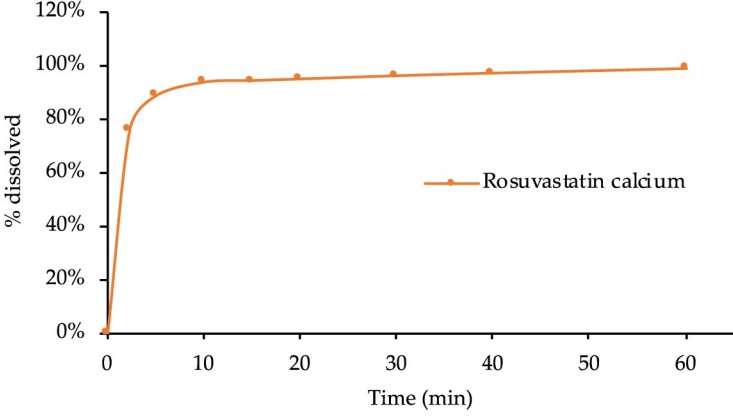

Fig 12. Dissolution rate of rosuvastatin calcium tablets.

33.34 N. The acceptance value (4.92) is within the limits allowed as established in the pharmacopoeia with an AV of less than 15. The acceptance value (4.92) in the contetn uniformity test is within the permitted limits as established by the pharmacopoeia with an AV less than 15. In turn, RFE 2.9.7 establishes that the percentage of weight loss must be less than 1%, the friability test complies with the specifications with a deviation of 0.40%. Finally, it is observed an excellent disintegration and dissolution profile of the tablets, the Fig 12 shows the high dissolution rate of the active ingredient with 93% of the drug dissolved after 10 minutes, being released practically in its totality at 30 minutes. The disintegration of six tablets in distilled water at a temperature of 37.0 ± 0.5 ˚C takes 9.00–9.48 seconds.

## 4. Conclusions

The use of amorphous rosuvastatin calcium with the selected excipients in the mixture did not showed any interactions between them and are suitable for their use in direct compression and

the tablets show hardness, disintegration, dissolution, and friability features that strictly meet pharmacopoeia requirements.

Direct compression of the above mixture resulted in resistant tablets with more than 90% release of the active substance in less than 10 minutes. These high values in such a short dissolution time can be attributed to the good solubility of the active ingredient, which could be increased by the presence of calcium phosphate, which by alkalizing the medium, favors the already good dissolution of the active ingredient.

On the other hand, the incorporation of superdisintegrants, such as crospovidone, provides higher water absorption and increased surface area, which leads to shorter disintegration times and, therefore, to an increase in the dissolution rate.

According to the SeDeM Expert System for the evaluation of rheological and particle size characteristics, the selected formulation possesses suitable properties for use in direct compression, the selected excipients, and their proportions, as well as the manufacturing method used in the development of this new rosuvastatin calcium tablets, are favorable and cost-effective.

The studies of characterization and compatibility drug-excipient by SEM, DSC, FT-IR and PXRD analysis have ensured an adequate adaptation in the different technological stages, always keeping quality as a primary objective, according to the philosophy of quality by design.

The mixture of amorphous calcium rosuvastatin and the selected excipients made it possible to obtain cost effective tablets by direct compression with optimal pharmacotechnical characteristics, which make it possible to foresee adequate stability and bioavailability, which must be verified by subsequent studies.

## Author Contributions

**Conceptualization:** Mª Ángeles Peña.

**Formal analysis:** Rocío González.

**Investigation:** Mª Ángeles Peña.

**Methodology:** Rocío González, Norma Sofía Torres, Guillermo Torrado.

**Supervision:** Mª Ángeles Peña, Guillermo Torrado.

**Validation:** Mª Ángeles Peña, Guillermo Torrado.

**Visualization:** Norma Sofía Torres.

**Writing – original draft:** Rocío González, Mª Ángeles Peña.

**Writing – review & editing:** Rocío González, Mª Ángeles Peña.

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
