## [Decision Letter · Decision Letter 0]

13 Jan 2022

PONE-D-21-36028Design, development, and characterization of amorphous rosuvastatin calcium tabletsPLOS ONE

Dear Dr. Peña Fernández,

Thank you for submitting your manuscript to PLOS ONE. After careful consideration, we feel that it has merit but does not fully meet PLOS ONE’s publication criteria as it currently stands. Therefore, we invite you to submit a revised version of the manuscript that addresses the points raised during the review process.

We look forward to receiving your revised manuscript.

Kind regards,

Il Won Kim

Academic Editor

PLOS ONE

Journal Requirements:

2. We noted in your submission details that a portion of your manuscript may have been presented or published elsewhere. ( DOI: 10.20944/preprints202111.0131.v1) Please clarify whether this publication was peer-reviewed and formally published. If this work was previously peer-reviewed and published, in the cover letter please provide the reason that this work does not constitute dual publication and should be included in the current manuscript.

Reviewers' comments:

Reviewer's Responses to Questions

**Comments to the Author**

1. Is the manuscript technically sound, and do the data support the conclusions?

Reviewer #1: Yes

Reviewer #2: Yes

2. Has the statistical analysis been performed appropriately and rigorously? 

Reviewer #1: Yes

Reviewer #2: Yes

3. Have the authors made all data underlying the findings in their manuscript fully available?

Reviewer #1: No

Reviewer #2: Yes

4. Is the manuscript presented in an intelligible fashion and written in standard English?

Reviewer #1: Yes

Reviewer #2: Yes

5. Review Comments to the Author

Reviewer #1: The using of SeDeM methodology is very useful especially in the preformulation stage.

The author needs to clarify some points

1. Regarding the r value, how it was calculated

2. Also, regarding the r value presented in table 2, I think that fig. 11 represent these values and not figure 3, please check

3. According, to USP, it is recommended to make a content uniformity for tablets containing less than 25 mg of active ingredients and calculates the acceptance value. Did the author follow this requirement, and if so, it is better to add the AV beside the weight variation data.

Reviewer #2: The manuscript PONE-D-21-36028 entitled Design, development, and characterization of amorphous rosuvastatin calcium tablets is a certainly interesting piece of research with a clear focus on the translational potential of the developed rosuvastatin tablets prepared by direct compresion.

The manuscript is overall written in a easy-to-read way and it is highly appreciated.

However, the introduction section is suggested to be slightly reduced for the sake of conciseness.

Apart from that, the results section should preferably be renamed as results and discussion (as results are partly discussed already there) so that current discussion can be renamed as conclusions.

Lastly, Pharmacopeia methods should rather be specified in the main body of the text than be referenced at the end of the article as any other citation, as in this view it is much clear which experiments have been conducted following Pharmacopeia's procedures.

6. PLOS authors have the option to publish the peer review history of their article (what does this mean?). If published, this will include your full peer review and any attached files.

Reviewer #1: No

Reviewer #2: **Yes: **Juan Aparicio-Blanco

---

## [Author Response · Author response to Decision Letter 0]

12 Feb 2022

The modifications of the article have been made as the reviewers very correctly indicated, in addition a new file with the figures is included, because a small modification has been made in one of them.

In the article the changes have been highlighted in yellow as indicated. The authors of this work appreciate all the suggestions that have undoubtedly improved the quality of our work, we hope that everything has been done according to your instructions.

---

## [Decision Letter · Decision Letter 1]

28 Feb 2022

Design, development, and characterization of amorphous rosuvastatin calcium tablets

PONE-D-21-36028R1

Dear Dr. Peña Fernández,

We’re pleased to inform you that your manuscript has been judged scientifically suitable for publication and will be formally accepted for publication once it meets all outstanding technical requirements.

Kind regards,

Il Won Kim

Academic Editor

PLOS ONE

Reviewers' comments:

Reviewer's Responses to Questions

**Comments to the Author**

1. If the authors have adequately addressed your comments raised in a previous round of review and you feel that this manuscript is now acceptable for publication, you may indicate that here to bypass the “Comments to the Author” section, enter your conflict of interest statement in the “Confidential to Editor” section, and submit your "Accept" recommendation.

Reviewer #1: All comments have been addressed

Reviewer #2: All comments have been addressed

2. Is the manuscript technically sound, and do the data support the conclusions?

Reviewer #1: Yes

Reviewer #2: Yes

3. Has the statistical analysis been performed appropriately and rigorously? 

Reviewer #1: Yes

Reviewer #2: Yes

4. Have the authors made all data underlying the findings in their manuscript fully available?

Reviewer #1: Yes

Reviewer #2: Yes

5. Is the manuscript presented in an intelligible fashion and written in standard English?

Reviewer #1: Yes

Reviewer #2: Yes

6. Review Comments to the Author

Reviewer #1: The manuscript was totally improved, all the missed and misunderstanding information now are clear. I suggest to accept the manuscript without any further modification.

Reviewer #2: (No Response)

7. PLOS authors have the option to publish the peer review history of their article (what does this mean?). If published, this will include your full peer review and any attached files.

Reviewer #1: No

Reviewer #2: Yes

---

## [Editor Report · Acceptance letter]

11 Mar 2022

PONE-D-21-36028R1 

Design, development, and characterization of amorphous rosuvastatin calcium tablets 

Dear Dr. Peña:

I'm pleased to inform you that your manuscript has been deemed suitable for publication in PLOS ONE. Congratulations! Your manuscript is now with our production department. 

Kind regards, 

on behalf of

Professor Il Won Kim 

Academic Editor

PLOS ONE